# Gravitational-wave asteroseismology with fundamental modes from compact binary inspirals

Geraint Pratten [1,2 ✉], Patricia Schmidt [1 ✉] & Tanja Hinderer [3,4 ✉]

Gravitational waves (GWs) from binary neutron stars encode unique information about ultra-dense matter through characterisic signatures associated with a variety of phenomena including tidal effects during the inspiral. The main tidal signature depends predominantly on the equation of state (EoS)-related tidal deformability parameter $\Lambda$, but at late times is also characterised by the frequency of the star's fundamental oscillation mode ($f$-mode). In General Relativity and for nuclear matter, $\Lambda$ and the $f$-modes are related by universal relations which may not hold for alternative theories of gravity or exotic matter. Independently measuring $\Lambda$ and the $f$-mode frequency enables tests of gravity and the nature of compact binaries. Here we present directly measured constraints on the $f$-mode frequencies of the companions of GW170817. We also show that future GW detector networks will measure $f$-mode frequencies to within tens of Hz, enabling precision GW asteroseismology with binary inspiral signals alone.

[1] School of Physics and Astronomy and Institute for Gravitational Wave Astronomy, University of Birmingham, Edgbaston, Birmingham B15 9TT, United Kingdom. [2] Universitat de les Illes Balears, Crta. Valldemossa km 7.5, E-07122 Palma, Spain. [3] GRAPPA, Anton Pannekoek Institute for Astronomy and Institute of High-Energy Physics, University of Amsterdam, Science Park 904, 1098 XH Amsterdam, The Netherlands. [4] Delta Institute for Theoretical Physics, Science Park 904, 1090 GL Amsterdam, The Netherlands. ✉email: gpratten@star.sr.bham.ac.uk; pschmidt@star.sr.bham.ac.uk; t.hinderer@uva.nl

The first detection of gravitational waves (GWs) from the binary neutron star (NS) inspiral GW170817[1] provided a distinctive avenue for probing ultra-dense matter and fundamental interactions in extreme regimes, at the frontier of nuclear physics and astrophysics. This observation yielded the first constraints on the yet unknown equation of state (EoS) of NS matter from the imprint of tidal interactions in the GW signal. Tidal signatures in GWs arise from the response of a matter object to the spacetime curvature sourced by its binary companion. They crucially depend on the EoS and are predominantly characterised by the tidal deformability parameters $\Lambda_\ell$[2], where $\ell = 2, 3$ denotes the quadrupole and octupole respectively. As the binary evolves towards merger, additional dynamical tidal effects become important when variations in the tidal fields occurring on the orbital timescale approach a resonance with the star's internal oscillation modes, giving rise to new characteristic signatures in the GWs associated with specific modes. Among these modes, the fundamental ($f_\ell$-)modes have the strongest tidal coupling and lead to a $f$-mode frequency-dependent amplification of tidal signatures in the GW signal that starts to accumulate long before the resonance[3–5]. In General Relativity (GR), and for a range of proposed nuclear EoSs for NSs, the $f_\ell$-mode frequencies are empirically found to be related to $\Lambda_\ell$ through approximate universal relations (URs)[6,7]. However, not imposing such URs opens up the possibility of using GW observations to perform parameterised tests of GR, to understand properties of matter at supranuclear densities such as possible phase transitions from hadronic to quark matter[8], and to test for the existence of exotic compact objects such as boson stars or gravastars[9].

In the past, prospects for $f$-modes asteroseismology with GW observations were focused on the post-merger signal[10–12]. With the observation of the inspiral of GW170817, recent work has substituted the constraints on the tidal deformability derived from GW170817 into empirical URs to indirectly make predictions for bounds on other NS parameters, such as the moment of inertia or $f$-mode frequencies. In ref. [13] bounds on $f$-modes were obtained by extrapolating a point estimate of the tidal deformability for GW170817 assuming URs.

Here we show direct constraints on $f$-modes from the GW data alone, without the assumption of URs, by using a model for the GW signal that explicitly depends both on the tidal deformability and the $f$-mode frequency[5]. In this UR-independent analysis, we find that the data from GW170817 disfavours anomalously small values of $f_\ell$. Our analysis enables tests of GR and exotic compact objects that would not be possible using point estimates and empirical URs. Our work thus demonstrates the concrete example of $f$-mode astereoseismology with GWs from binary inspirals, opening up this highly anticipated method for future detailed probes of compact object interiors.

## Results

### $f$-mode constraints from GW170817.
In order to constrain fundamental oscillation mode frequencies in GW170817, we re-analyse the publicly available strain data[14,15] treating the tidal deformabilities $\Lambda_{\ell,A}$ and the dimensionless angular $f$-mode frequencies $\Omega_{\ell,A} \equiv Gm_A\omega_{\ell,A}/c^3$ of the A-th component object as independent parameters without imposing the aforementioned empirical URs or any requirement that the two compact objects obey the same EoSs. We then repeat the analysis imposing URs that relate the $f$-mode frequency and the tidal deformability, $\Omega_{\ell,A} = \Sigma_i a_i \xi^i$, where $\xi = \ln(\Lambda_{\ell,A})$ and $a_i$ are numerical coefficients[6]. For our analyses, we use the efficient post-Newtonian (PN) model of the frequency-domain waveform "TaylorF2" that includes adiabatic tidal effects (see references in ref. [16]) augmented with the dynamical $f$-mode tidal contribution to the

**Table 1 Marginalised 90% lower bound or 90% credible interval of the quadrupolar $f$-mode frequencies in GW170817.**

| Tidal phase model | $f_{2,1}$ [kHz] | $f_{2,2}$ [kHz] |
|---|---|---|
| (a) 6.5PN ad. + $f_2$ dyn. | 1.47 | 1.57 |
| (b) 7.5PN ad. + $f_2$ dyn. | 1.43 | 1.59 |
| (c) Combined | 1.45 | 1.58 |
| (d) 6.5PN ad. + $f_2 + f_3$ dyn. | 1.40 | 1.49 |
| (e) 7.5PN ad. + $f_2 + f_3$ dyn. | 1.37 | 1.47 |
| (f) Combined | 1.39 | 1.48 |
| (g) 6.5PN ad. + URs | 1.36–2.83 | 1.42–3.08 |
| (h) 7.5PN ad. + URs | 1.37–2.90 | 1.43–3.16 |
| (i) Combined | 1.38–2.86 | 1.43–3.12 |
| (j) 6.5PN ad. + $f_2$ dyn. + URs | 1.42–2.88 | 1.48–3.17 |
| (k) 7.5PN ad. + $f_2 + f_3$ dyn. + URs | 1.44–2.92 | 1.50–3.18 |
| (l) Combined | 1.43–2.90 | 1.48–3.18 |

We list results for all considered combinations of adiabatic and dynamical tide contributions in the waveform model for both companions of GW170817 assuming a uniform prior on $\Omega_{2,A} \in [0, 0.5]$. For results which assume URs, we also provide the upper bounds. Fig. 1b displays cases (f), (i) and (l). Cases where $f_3$ is included also include the 7PN adiabatic octupolar effects[52]. The results in this table demonstrate that the data disfavours hyper-excited dynamical tides independent of PN systematics.

phase developed in ref. [5] (see Methods, Waveform Model). We perform coherent Bayesian inference using the nested sampling algorithm implemented in LALINFERENCE[17,18], as summarised in Methods, Parameter Estimation. We adopt the narrow spin priors of[16], limiting the dimensionless neutron star spins to be $\leq 0.05$, which is consistent with the observed population of binary neutron stars that will merge within a Hubble time[16]. For the $f$-mode frequencies, we adopt an agnostic uniform prior. The choice of priors is further discussed in Methods, Choice of Priors and Parameterisation. We further demonstrate the robustness of our results under systematic uncertainties from incomplete knowledge of PN corrections to adiabatic tidal effects by performing the analysis using different combinations of adiabatic and dynamical tide contributions, as summarised in Table 1. The results presented in the text below will be taken from datasets (f) and (l) unless stated otherwise.

Figure 1 summarises our findings for the larger-mass (labelled here as object A = 1 with mass $m_1$) companion of GW170817; we obtain similar results for the lower-mass ($m_2$) companion resulting in the same conclusions as illustrated in Supplementary Fig. 1. Figure 1a shows the joint posterior probability distribution function (PDF) for the quadrupolar $f$-mode frequency $f_2 = \omega/(2\pi)$ and corresponding tidal deformability $\Lambda_2$. We recall that in the subscripts $\{2, 1\}$ the first label denotes the multipolar index $\ell$, the second specifies the object label A, with A = 1 taken to be the larger mass companion here. The black solid curve indicates the 90% credible region (CR) of our analysis with independent parameters, i.e., without imposing the UR. We find that the data disfavours large tidal deformabilities and small $f$-mode frequencies. Complementary to the joint two-dimensional posterior, Fig. 1b shows the marginalised one-dimensional PDF of $f_2$. It is evident that without imposing the UR for NSs on the dynamical tides terms (green curve), we can rule out anomalously small values of the $f$-mode frequency and place a lower bound on $f_2$. This can be understood from the scaling behaviour of the $f_\ell$-dependent phase contribution which is proportional to $\Lambda_\ell f_\ell^{-25}$, implying that small values of $f_\ell$ result in hyper-excited dynamical tides that are inconsistent with the data. This is reflected in the shape of the posterior (green curve), where we see a dramatic drop in posterior support as $f \to 0$. Conversely, increasing the fundamental frequency leads to a suppression of dynamical tidal effects and the waveform becomes indistinguishable from the

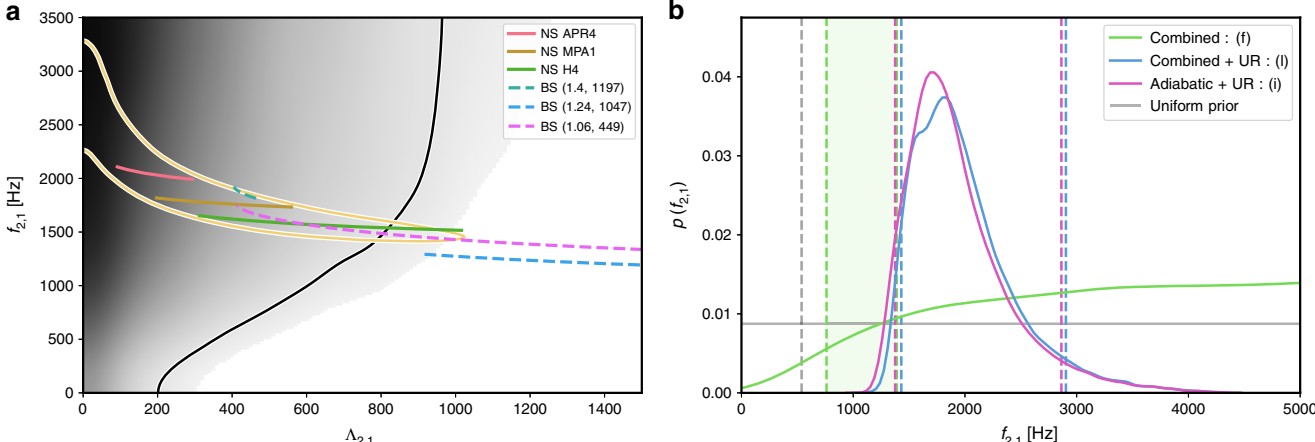

**Fig. 1 Results for the larger-mass object of GW170817. a** The black shaded region shows the two-dimensional probability distribution function (PDF) for the $f_2$-mode frequency and quadrupolar tidal deformability of the larger companion $\Lambda_{2,1}$, where the subscripts denote the multipolar index $\ell = 2$ and the larger mass object's label A = 1. The solid lines correspond to the 90% credible regions, where the black curve corresponds to the analysis in which $f_{2,A}$ are treated as independent parameters and the yellow one to imposing the universal relations, i.e., fixing $f_{2,A}$ given $\Lambda_{2,A}$. The posteriors are overlaid with UR predictions for three EoS for NSs (coloured solid curves), and three massive BSs (coloured dashed curves) denoted $(m_b/m_n, \lambda_b)$, with $m_n = 1.675 \times 10^{-27}$ kg being the neutron mass, where all curves are restricted to the 90% interval of the component mass posterior, $m_1 \in [1.37, 1.63]$ M$_\odot$. **b** Marginalised one-dimensional PDF for the $f_2$-mode frequency. We show results for the following three tidal phase models as listed in Table 1: (i) purely adiabatic tides with URs imposed (pink), (l) adiabatic and dynamical tides with UR imposed (blue) and (f) adiabatic and dynamical tides without UR assumed (green). The dashed lines indicate the corresponding 90% lower bound (green) for the UR-independent analysis or credible interval (pink and blue) for the results with the UR imposed. The green shaded region indicates how the lower bound changes in the UR-independent analysis when a different upper limit on the prior for $\Omega_{2,A}$, ranging between 0.182 and 0.5, is assumed.

adiabatic limit. The dynamical $f$-mode effects become most important at high frequencies where the detectors are less sensitive, thus the $f$-mode frequency could not be fully resolved and no upper bound could be determined as seen from the plateau at high frequencies. This observation motivates gravitational-wave detectors with sufficient sensitivity at high frequencies to distinguish between the adiabatic and dynamical tidal contributions, as highlighted in the next section. Depending on the choice of the upper prior limit for $\Omega_{2,A}$, the value of the lower posterior bound changes, which we indicate by the shaded green region. The lower boundary of the green shaded region corresponds to choosing an upper prior limit of $\Omega_{2,A} = 0.182$, the UR limit for $\Lambda_{2,A} \to 0$. The upper boundary of the green shaded region corresponds to choosing a conservative upper limit on the prior of $\Omega_{2,A} = 0.5$. The sensitivity of the posteriors to the prior is due to our inability to distinguish between the adiabatic and dynamical tidal contributions for large $f$-mode frequencies at current detector sensitivities. We note, however, that there are no known realistic EoS that predict such high $f$-mode frequencies.

In addition, we also show the two-dimensional (Fig. 1a, yellow curve) and one-dimensional (Fig. 1b, blue curve) posteriors obtained by imposing URs for NSs. We find that the results are consistent with our UR-independent analysis and have, by construction, narrower bounds. In Fig. 1b we also show the results for a purely adiabatic tidal phase model (pink curve) for completeness. This entirely omits the frequency-dependent dynamical tidal enhancement from the GW model and reconstructs the $f$-mode posteriors from the tidal deformabilities via the URs. We find that the lower bound is consistent with our estimates using the dynamical tides model—this is not surprising as the strongest constraint stems from the tidal deformability measurement, which would be inconsistent with very large dynamical tides (i.e., $f_2 \to 0$).

We emphasise that the PDFs derived by imposing URs are only valid for neutron stars. In general, exotic compact objects and objects in modified theories of gravity may not obey the same set of URs, and even for NSs within GR, the appearance of new states of QCD matter in the high-density cores can deteriorate the accuracy of URs, highlighting the utility of our approach. Figure 1 illustrates that measuring both the $f$-mode frequency and $\Lambda$ independently enables us to place additional constraints on the nature of the compact objects. As an example of this concept, the coloured solid curves correspond to predictions from EoS models for NSs with increasing stiffness known as APR4[19], MPA1[20], and H4[21] where the $f$-mode was calculated from URs; as an example of an exotic object we also show predictions for nonrotating boson stars (dashed coloured curves)[22,23]. The $f$-mode frequencies computed in ref. [23] use an approximate effective EoS and are used here solely for illustration as more complete results for cases relevant here are not available in the literature. Boson stars (BSs) are condensates of a complex scalar field $\Phi$ with a repulsive self-interaction described here by the potential $V = m_b^2 |\Phi|^2 + \frac{1}{2}\lambda_b |\Phi|^4$, where $m_b$ is the boson mass and $\lambda_b$ characterises the strength of the self-interaction.

While not possible for GW170817, a highly localised posterior in the $f$-$\Lambda$ plane will afford us the ability to distinguish between different EoS and perform tests of exotic compact objects and modified theories of gravity in future GW observations.

The bounds on $f$-modes presented in ref. [13], obtained by extrapolating point estimates, are in broad agreement with the results in our analysis. We stress, however, that such estimates are based on empirical URs for nuclear EoSs and only apply to a canonical $1.4 M_\odot$ NS, significantly limiting the analysis. The fully Bayesian inference presented here yields the full multidimensional posteriors, providing information on the correlations between the masses, tidal deformabilities and $f$-mode frequencies observed in GW170817 (see Supplementary Fig. 2). This information is vital for performing tests of GR and placing constraints on exotic compact objects. Such tests are not possible using the methods of ref. [13].

In Table 1 we give the 90% lower bounds on the quadrupolar $f$-mode frequency measured from GW170817 for various combinations of adiabatic and dynamical tidal phase contributions with and without assuming URs. The analysis assumes a uniform prior on $\Omega_{2,A} \in [0, 0.5]$. We follow[24] and construct the bootstrapping estimate of the standard error on the lower limit, finding $f_{2,1}^{\text{low}} = 1390 \pm 65$ Hz for the larger mass. In particular, we note that the systematics between the different PN contributions are all within the bootstrapping errors, demonstrating robustness of the result. We caution, however, that for GW170817, the lower bound on the $f$-mode frequency is sensitive to the upper prior bound, as demonstrated in Fig. 1.

In addition to the dominant quadrupole effects, we can also place very weak constraints on the octupolar tidal parameters (see Supplementary Fig. 3). Such higher multipole tidal interactions are subdominant and hence even more difficult to constrain from the data. For the larger component, the 90% credible interval on $\Lambda_{3,1}$ is [410, 9404] and the 90% lower bound on $f_{3,1}^{\text{low}}$ is $1857 \pm 117$ Hz. Similarly, for the smaller component we find $\Lambda_{3,2} \in [466, 9446]$ and $f_{3,2}^{\text{low}} = 1619 \pm 98$ Hz. Note that the limits on $\Lambda_{3,A}$ are prior-dominated whereas the data also disfavour hyper-excited octupolar dynamical tides.

While our analysis enables us to place constraints on the $f$-modes for GW170817 from below and rule out very low values of $f_\ell$, little information is gained overall due to the decreased sensitivity at high frequencies for the current GW detector network. However, future detector networks will have the potential to place substantially tighter constraints on fundamental modes as we demonstrate next.

**Measurement prospects in future detector networks.** To illustrate the feasibility of measuring $f$-modes from inspiral signals with future GW observatories, we consider a GW170817-like binary with component masses $m_A = \{1.475, 1.26\}$ $M_\odot$, tidal deformabilities $\Lambda_{2,A} = \{183, 488\}$ and $f_2$-mode frequencies $f_{2,A} = \{2.04, 1.94\}$ kHz based on the APR4 EoS[19]. At high signal-to-noise ratios (SNRs), we use the linear expansion of the GW signal $h$ as a function of the binary parameters $\vec{\lambda}$ around the true parameter values $\vec{\lambda}_0$ given by $h(\vec{\lambda}) = h(\vec{\lambda}_0) + \partial_i h \Delta \lambda^i + O(\Delta \lambda^2)$, where $\Delta \lambda^i = \vec{\lambda} - \vec{\lambda}_0$, to approximate the multivariate PDF of $\vec{\lambda}$ about $\vec{\lambda}_0$ as $p(\vec{\lambda}_0) \sim e^{-(1-\mathcal{M})\rho^2}$[25,26]. Here $\rho$ is the SNR and $1 - \mathcal{M}$ is the mismatch between two waveforms, where $\mathcal{M}$ is the normalised, noise-weighted inner product optimised over time and phase shifts (see Eq. (3)). In our analysis here we use for $h$ the TaylorF2 approximant with 6.5PN adiabatic tidal effects and only include the $f_2$ dynamical tides contribution; we do not impose URs. Figure. 2 shows the approximate one-dimensional PDF for the $f_2$-mode frequency of the larger-mass object for an optimally-oriented GW170817-like binary at 40 Mpc in different detector networks. The result for the LIGO-Virgo detector network at design sensitivity[31,32] shows little improvement over our actual measurement for GW170817 (compare to Fig. 1b), only allowing us to determine a lower bound. However, various future network configurations all enable much tighter constraints on the $f$-mode frequencies, highlighting the potential of such detectors: A network of three A+ detectors[27,28] will offer significant improvements over the Advanced LIGO-Virgo (HLV) network. In particular, we find that we can begin to distinguish between an adiabatic waveform and dynamical tides sourced by large $f$-mode frequencies. By optimising current detectors at high frequencies[29,30] (HF4S), we find that we can start to place meaningful 90% lower and upper limits on $f_2$ (green curve). Given the high SNRs anticipated in a future 3G network (see

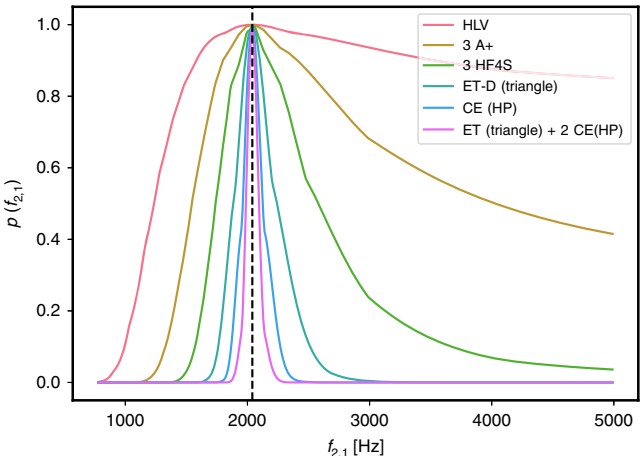

**Fig. 2 1D posterior probability for the $f_2$-mode frequency of a GW170817-like binary in different detector networks.** The vertical dashed line indicates the true value of $f_2 = 2.04$ kHz for the larger mass. The detector networks considered are: LIGO-Virgo at design sensitivity (HLV), three A+ detectors[28,29], three 4 km L-shaped LIGO detectors with improved high-frequency sensitivity[30,31] (HF4S), one triangular Einstein Telescope D-configuration, one Cosmic Explorer (CE) and a network consisting of two CEs and one ET-D.

Supplementary Table I), we can estimate the statistical accuracy to which we expect to be able to measure model parameters through a Fisher analysis. A network consisting of one Einstein Telescope (ET) detector[33,34] and two Cosmic Explorer (CE) observatories[35] shows great overall promise for enabling precision measurements of the binary parameters: with the component masses being measurable to the sub-percent level and tidal deformabilities being measurable to within a few percent. Crucially, it will also allow for $1\sigma$-errors of the $f$-mode frequencies of only a few tens of Hz, making precision GW asteroseismology with inspiral signals possible.

This simplified calculation neglects correlations between intrinsic parameters and is limited by systematics of the waveform model; it should be viewed as a proof of principle that we can make meaningful measurements of fundamental mode oscillations from compact binary inspirals. A detailed study on the measurability of $f$-modes in future GW detector networks will be presented in forthcoming work.

**Discussion**

We present direct constraints on the fundamental oscillation modes in GW170817 using a waveform model with explicit dynamical tides and without assuming URs, demonstrating how we can directly measure the fundamental oscillation mode frequencies from the GW data alone. The implications of these results are: (i) hyper-excited dynamical tides, i.e., anomalously small $f$-mode frequencies, are disfavoured by GW170817, (ii) current GW detections only allow for lower bounds to be placed on the quadrupolar and octupolar $f$-mode frequencies, (iii) the lower bounds are consistent with the predictions from URs for nuclear EoSs but do depend on the choice of the upper limit of the prior. We have further demonstrated that meaningful measurements of $f$-mode dynamical tides are a unique possibility in a future GW detector network. Such networks will be able to measure the masses and tidal deformabilities to high accuracy, and the $f$-mode frequency to within tens of Hz without the assumption of URs. Precise measurements of the $f$-mode frequency for populations of sources will open up the possibility of probing the transition to quark matter in NS cores, chartering an

unexplored regime in the QCD phase diagram through complementary information than is contained in possible GW signatures from other oscillation modes. Moreover, the highly localised PDF in the $\Lambda$-$f$ plane anticipated with the future networks will help break degeneracies and confront the predictions from URs with stringent observational tests, enabling to discriminate between exotic compact objects and conventional nuclear matter models. In addition, it will help constrain alternative theories of gravity and exotic physics beyond the standard model such as axions.

While we have only considered quasicircular binaries here, binaries with non-negligible eccentricity may have even larger excitations of dynamical $f$-mode tides[36–39]. Though no significant BNS candidates with non-negligible eccentricities have been found in GW searches to date[40].

Finally, we wish to highlight the potential complementarity between GW asteroseismology in the inspiral and information obtained from the post-merger signal[11,12]. The frequencies emitted during the post-merger phase depend on the properties of the merger remnant and therefore depend on the mass, spin and EoS of the progenitor binary. The pre-merger phase is a probe of the cold EoS, whereas the post-merger phase probes the hot EoS at densities several times larger than the nuclear saturation density. By accurately measuring the $f$-mode frequency and mass of a neutron star, we can infer the moment-of-inertia $I$ and hence the radius of the star, so long as the EoS does not exhibit significant softening at super-nuclear densities[6,41,42]. This would enable us to probe matter at densities well above the nuclear saturation density[41], making gravitational-wave asteroseismology a potentially powerful tool in constraining the EoS of compact objects and probing the physics of their interiors. Combined with post-merger constraints on the hot EoS, we can begin to probe the behaviour of the EoS under a range of astrophysical conditions. In addition, independent measurements of the $f$-mode frequency, the tidal deformability and the mass of the NS would enable constraints on the effective no-hair relations for neutron stars and quark stars[6,43].

The analysis and methods in this paper thus lay the foundation for opening unique prospects for deriving fundamental information from compact binary inspirals.

## Methods

### Parameter estimation.
We perform coherent Bayesian parameter estimation on the publicly available GW170817 data[14,15]. We use the nested sampling algorithm implemented in LALINFERENCE[18,44], part of the publicly available LIGO Algorithms Library (LAL)[45], to evaluate the posterior probability distribution

$$p(\vec{\lambda}|\vec{d}) = \frac{\mathcal{L}(d|\vec{\lambda})\pi(\vec{\lambda})}{\mathcal{Z}(\vec{d})}, \tag{1}$$

where $\mathcal{L}(\vec{d}|\vec{\lambda})$ is the likelihood of the data given the parameters $\vec{\lambda}$, $\pi(\vec{\lambda})$ the prior distribution for $\vec{\lambda}$ and $\mathcal{Z}$ the marginalised likelihood or evidence. Under the assumption of stationary Gaussian noise, the likelihood of obtaining a signal $h$ in the data $\vec{d}$ is given by

$$\mathcal{L}(\vec{d}|\vec{\lambda}) \propto \exp\left[-\frac{1}{2}\sum_k \langle h_k^M(\vec{\lambda}) - d_k | h_k^M(\vec{\lambda}) - d_k \rangle \right], \tag{2}$$

where the $k$-th detector output is $d_k(t) = n_k(t) + h_k^M(t; \vec{\lambda})$, $n_k(t)$ the noise and $h_k^M(t)$ the measured strain incorporating calibration uncertainty[46]. For a single detector, the noise weighted inner product is given by

$$\langle a|b \rangle = 4\mathrm{Re}\int_{f_{\mathrm{low}}}^{f_{\mathrm{high}}} \frac{\tilde{a}(f)\tilde{b}^*(f)}{S_n(f)}\, df, \tag{3}$$

where $\tilde{a}(f)$ and $\tilde{b}(f)$ denote the Fourier transform of the real-valued functions $a(t)$ and $b(t)$ respectively and $S_n(f)$ is the power spectral density (PSD) of the detector.

In our analysis we adopt a low-frequency cutoff of $f_{\mathrm{low}} = 23$ Hz and a high-frequency cut-off of $f_{\mathrm{high}} = 2048$ Hz.

### Choice of priors and parameterisation.
The choice of priors used in our analysis is templated on the choices made in ref. [46]. For the dimensionless tidal deformabilities we use uniform priors with $\Lambda_{2,\mathrm{A}}^{\mathrm{prior}} \in [0, 5000]$ and $\Lambda_{3,\mathrm{A}}^{\mathrm{prior}} \in [0, 10^4]$. Similarly, where appropriate, we adopt uniform priors on the dimensionless angular $f$-mode frequencies $\Omega_{\ell,\mathrm{A}} = Gm_\mathrm{A}\omega_{\ell,\mathrm{A}}/c^3$. We consider a range of upper values on $\Omega_{2,\mathrm{A}}$ in order to gauge the impact of the prior on the $f$-mode bounds, as discussed in the text. Our default prior is taken to be $\Omega_{2,\mathrm{A}} \in [0, 0.5]$ and $\Omega_{3,\mathrm{A}} \in [0, 1.0]$. These priors are decidedly conservative in order to remain as agnostic as possible. The upper limit imposed on $\Omega_{\ell,\mathrm{A}}$ extends beyond the limits of ~0.18 ($\ell = 2$) and ~0.22 ($\ell = 3$) implied by URs as $\Lambda_{\ell,\mathrm{A}} \to 0$[6].

In addition, there is some freedom associated to how we sample in the intrinsic parameters. Following[16], we let the adiabatic tidal parameters $\Lambda_{2,\mathrm{A}}$ vary independently and considered two different prescriptions for incorporating dynamical tides. In the first prescription, we let the $f$-mode frequencies vary independently, being treated as a free parameter to be constrained by the data. This allows us to place a lower bound on the $f$-modes avoiding any a priori assumptions on the validity of the universal relations. In the second prescription, we fix the $f$-mode frequencies by imposing empirical universal relations, though we note that the dynamical tides terms still explicitly contribute to the log-likelihood in contrast to a purely adiabatic waveform. Imposing universal relations allows us to effectively reduce the dimensionality of the parameter space whilst still incorporating dynamical tidal effects.

### Waveform model.
In this study, we analysed GW170817 data using a fixed point particle baseline $\Psi_{\mathrm{pp}}$ (see ref. [16] and references therein for details) but incorporating adiabatic and dynamical tidal effects to varying PN orders. The aim of this approach is to gauge the impact of systematics in the modelling of tidal effects on the estimation and bounds for the $f$-mode frequencies.

The leading order adiabatic tidal effects first enter the phase at 5PN order[2] and can be characterised by a single tidal deformability parameter $\tilde{\Lambda}$, which is a mass weighted average of the tidal deformabilities of the constituent compact objects $\Lambda_{2,\mathrm{A}}$. The effects of adiabatic tidal deformations on the phase have been calculated to 6PN in ref. [47] and up to 7.5PN in ref. [48], where contributions from higher multipoles are omitted and a number of unknown terms appearing at 7PN are neglected. In this paper, we also incorporate a recently derived closed-form expression for the dynamical tidal contribution to the phase[5]. This enables us to include the tidal excitation of the neutron stars fundamental oscillation modes at quadrupolar ($\ell = 2$) and octupolar ($\ell = 3$) order. In addition, we also include the octupolar adiabatic contribution to the phase as in ref. . The GW phase $\Psi$ computed in the PN approximation can therefore be written as the sum of the various contributions

$$\Psi(f) = \Psi_{\mathrm{pp}}(f) + \Psi_{\mathrm{ad.}}(f) + \Psi_{\mathrm{dyn.}}(f), \tag{4}$$

where $\Psi_{\mathrm{pp}}$ is the point particle inspiral phase, $\Psi_{\mathrm{ad.}}$ the adiabatic tidal contributions to the phase and $\Psi_{\mathrm{dyn.}}$ the dynamical tidal contributions to the phase (fmtidal) from ref. [5]. We neglect quadrupole-monopole (QM) effects associated with the deformations of the NS under its own angular momentum[49], as for GW170817 the QM effects were found to be subdominant[50]. However, the QM term can readily be incorporated into our phasing model. Future studies could further include a number of additional matter effects neglected here, and use more sophisticated waveform models beyond the PN-based approximations.

Following[50], we choose the waveform termination frequency as the minimum of min ($f_{\mathrm{ISCO}}, f_{\mathrm{contact}}$), where $f_{\mathrm{ISCO}}$ is the innermost stable circular orbit and $f_{\mathrm{contact}}$ is a fiducial contact frequency.

Here, we have made use of a post-Newtonian waveform model, though we note that the phase model for dynamical tides[5] can be applied to any frequency domain waveform model, including models calibrated to numerical relativity simulations. Such calibrated waveform models yield a more accurate description of the point-particle phase, potentially reducing systematic errors or parameter biases. A more detailed comparison between the dynamical tidal model used here and other tidal waveform approximants is given in ref. [5]. However, a systematic study is beyond the scope of this paper.

## Data availability
The posterior samples and 1D PDFs that support the findings of this study are available in GWAsteroseismologyFModesDataRelease with the identifier https://doi.org/10.5281/zenodo.363493[51].

## Code availability
The analysis makes use of the publicly available LIGO Scientific Collaboration Algorithm Library Suite (LALSuite)[45].

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

## Acknowledgements

The authors thank Alberto Vecchio, Ben Farr and Guy Davies for useful discussions and comments on the manuscript, and Denis Martynov and Haixing Miao for providing us with the sensitivity curve used in Fig. 2. G.P. acknowledges support from the Spanish Ministry of Culture and Sport grant FPU15/03344, the Spanish Ministry of Economy and Competitiveness grants FPA2016-76821-P, the Agencia estatal de Investigación, the RED CONSOLIDER CPAN FPA2017-90687-REDC, RED CONSOLIDER MULTIDARK: Multimessenger Approach for Dark Matter Detection, FPA2017-90566-REDC, Red nacional de astropartículas (RENATA), FPA2015-68783-REDT, European Union FEDER funds, Vicepresidència i Conselleria d'Innovació, Recerca i Turisme, Conselleria d'Educació, i Universitats del Govern de les Illes Balears i Fons Social Europeu, Grav-itational waves, black holes and fundamental physics. P.S. acknowledges support from the Netherlands Organisation for Scientific Research (NWO) Veni grant no. 680-47-460. T.H. acknowledges support from the DeltaITP and NWO Projectruimte grant GW-EM NS. This research has made use of data, software and/or web tools obtained from the Gravitational Wave Open Science Center[14], a service of LIGO Laboratory, the LIGO Scientific Collaboration and the Virgo Collaboration. LIGO is funded by the U.S. National Science Foundation. Virgo is funded by the French Centre National de Recherche Scientifique (CNRS), the Italian Istituto Nazionale della Fisica Nucleare (INFN) and the Dutch Nikhef, with contributions by Polish and Hungarian institutes. The authors are grateful for computational resources provided by the LIGO Laboratory—Caltech Computing Cluster and supported by the National Science Foundation.

## Author contributions

G.P. led the implementation of the waveform model, enabling this study and subsequent parameter estimation analyses. P.S. was also actively involved in the model

implementation and parameter estimation results. P.S. and T.H. developed the waveform model employed in this analysis. T.H. produced the comparisons to boson star equations of state. All authors contributed to the underlying theoretical insight in the paper and to writing the manuscript.

## Competing interests

The authors declare no competing interests.
