## [Peer Review File · Nature Communications]

Reviewers' comments:

Reviewer #1 (Remarks to the Author):

This manuscript deals with the constraints on the fundamental modes (f-mode) in GW170817 by using a waveform model with explicit dynamical tides and without assuming universal relations and provides a way to measure the frequency of f-mode from the GW data.

As we know, extracting the information of oscillation modes from the GW radiated from the binary neutron star inspiral is a very meaningful but not easy task. This work obtained valued measured constraints on the f-mode frequencies of the companions of GW170817, which is very interesting and also meaningful in understanding the future detection of GW radiated from the binary neutron star inspiral.

I believe that the methods and the results in this work will be very useful in the studying of compact stars and dense matter and thus I recommend the publication of the article.

COMMENTS/SUGGESTIONS

(1) There are some recent works also focused on the f-mode excited in the last stage of the orbiting binary neutron stars, such as PRD 100 (2019) 063001, PRD 100 (2019) 064023, arXiv:1905.00012v1. I recommend that the authors could include in their discussion with these relevant papers.

(2) It is not easy for the readers to understand that which URs are adopted and how the URs get involved in the posteriors.

(3) Please check whether the dimensions of relevant quantities are consistent: in the penultimate paragraph of the right column in page 3, $\Lambda(3,1)$ is [410, 9404] Hz, $\Lambda(3,2)$ is [466, 9446] Hz; while in the last paragraph of the left column in page 5, $\Lambda(\text{prior}; 3,1)$ is dimensionless.

(4) The symbols of the quantities, such as $\Lambda(2,1)$ and $\Lambda(2,A)$, $f(2,1)$ and $f(2,A)$, $\Lambda(2,A)$ and $\Lambda(2A)$, etc. are easily confused, it is better to explain these symbols in a little more detail.

I believe that the above comments/suggestions should be taken into account and answered in the revised version of the article.

Reviewer #2 (Remarks to the Author):

The authors perform for the first time (and without assuming the so-called universal relations) a Bayesian study on the data of GW170817 to estimate frequencies of the f-modes excited during the final part of the merger. Separately from GW170817, such f-modes had also been studied in previous works, mentioned in the manuscript.

I believe that the results presented in this manuscript are technically sound and expressed in a easy-to-read and convincing manner (actually there are some repetitions, therefore the text could be shortened somewhat, if necessary). For sure these results are of interest to other researchers in the field, but I do not believe that they represent an advance in understanding likely to influence thinking in the field. In fact, this manuscript shows an application of known techniques and theory to some actual data. This is why I do not recommend publication in Nature Communications. I would definitely recommend publication in other types of journals.

Reviewer #3 (Remarks to the Author):

Dear Editor, dear Authors,

I read the manuscript "GW Astroseismology with Fundamental Modes from Compact Binary Inspirals" and I provide

below my comments about it.

The manuscript deals with a detailed Bayesian analysis of the GW strain data measured on the GW170817 event, by using a model for the GW signal that explicitly depends on both the tidal deformability and the f-mode frequency, assumed as independent variables. Differently from previous analysis, no empirical universal relations (URs) are necessarily assumed and a first constraint on the f-mode frequency only based on the available data is deduced. Moreover, a simple but clear test is done to reveal how this kind of analysis can improve our understanding of f-modes in future, better resolved detection.

The work is original and has a fairly good significance. Despite the fact that the results obtained are still compatible with the (possibly more constrained) bounds obtained by additionally imposing URs, this analysis has the potential advantage of revealing systematic uncertainties and of testing some of the hypothesis underlying the empirical URs. Moreover, future analysis of this kind, applied to a larger sample of good resolved GW events, can potentially reveal new phases of matter at very large supra-nuclear densities or test alternative theories of gravity.

The results presented can be of interest for a large (even if not extremely large) community. The Authors did an appreciable attempt to put their results in a broader and more understandable contest. However, the work has an intrinsic degree of technicality that cannot be avoided. Still, I do not think that this is a severe limitation of the work.

The data and the methodology are state-of-the-art and the authors are recognized experts in the field of GW data analysis. The references and the provided details are enough to enable reproducing the results. Bayesian analysis strongly depends, among the others, on the priors used in the analysis. The authors have investigated in a satisfactory way the possible dependence of their results on the priors on the f-mode frequency. The study presented in the second part

of the paper is less accurate, but being a proof-of-principle, this is not necessarily a limitation.

The conclusions reflect the main points of the paper. Some of the outlooks are not fully motivated, but it should be possible to amend.

Below I list a few possible suggestions to improve the manuscript:

1) I am not convinced by the switch between the method and the results sections. Some of the questions I had while reading the manuscript were (partially) answered in the method section. So, I am wondering if a more canonical ordering could be better suited.

2) In a few places in the paper, the Authors speak about ultra-high densities. Since the journal reader target is very broad, I would avoid this too general statements and I would replace them with more precise statement. At which densities do the authors expect their method to reveal matter properties (for example, in units of nuclear saturation density)?

3) The Authors used TaylorF2 waveform models. This is not the only available one, even if it has some advantages. Could the Authors comments on possible pro and cons with respect to other approximants, and if they expect possible differences using different approximants?

4) When commenting on "the f_l -dependent phase contribution (which is proportional to $\Lambda_l f_l^{-2}$)", a reference could be useful.

5) Very minor: sometimes (more often in the second part of the paper) the Authors do not put a comma in subscripts (e.g. Ω_{2A}), while I think it should be done (e.g. $\Omega_{2,A}$).

6) Very minor: after eq 3, "we adopt a a low- ..."

7) I do not fully understand why the lower bound on Ω is sometimes 0 and sometimes 0.182 (the lower limit of the URs). Is it not possible to have a single choice? Is there something that I am missing in using two different values?

8) When discussing the potential relevance of this kind of analysis in future detection, the Authors generically say that it will be complementary to post-merger analysis. There is however a possible fundamental difference: in the post-merger, the hot nuclear EOS will be probed, while in the inspiral the cold one. Could the Authors expand a little bit on that? Moreover, how does the SNR necessary in the presented analysis relates with the SNR required by post-merger analysis? This could be a relevant term of comparison.

We thank the referees for their insightful and constructive comments on the manuscript. We thank referee 1 for highlighting the “very interesting and meaningful nature of our work” and that the “methods and results in this work will be very useful” and therefore for recommending publication. We would also like to thank referee 3 for highlighting that the work is “original and has fairly good significance” with “the potential advantage of revealing systematic uncertainties and of testing some of the hypothesis underlying” universal relations. Referee 3 also points out that future analysis following our results can “potentially reveal new phases of matter at very large supra-nuclear densities or test alternative theories of gravity”. We have responded to each referee separately and have marked all changes in the manuscript as requested.

Reviewer #1 (Remarks to the Author):

This manuscript deals with the constraints on the fundamental modes (f-mode) in GW170817 by using a waveform model with explicit dynamical tides and without assuming universal relations and provides a way to measure the frequency of f-mode from the GW data. As we know, extracting the information of oscillation modes from the GW radiated from the binary neutron star inspiral is a very meaningful but not easy task. This work obtained valued measured constraints on the f-mode frequencies of the companions of GW170817, which is very interesting and also meaningful in understanding the future detection of GW radiated from the binary neutron star inspiral.

I believe that the methods and the results in this work will be very useful in the studying of compact stars and dense matter and thus I recommend the publication of the article.

COMMENTS/SUGGESTIONS

(1) There are some recent works also focused on the f-mode excited in the last stage of the orbiting binary neutron stars, such as PRD 100 (2019) 063001, PRD 100 (2019) 064023, arXiv:1905.00012v1. I recommend that the authors could include in their discussion with these relevant papers.

- We have added the references. References PRD 100 (2019) 063001 and PRD 100 (2019) 064023 specialize to the context of eccentric inspirals, and have been added in our discussion of such scenarios. The preprint by Andersson and Pnigouras investigates the relations between f-modes and Love numbers in a Newtonian context and has been added to that discussion in the text.

(2) It is not easy for the readers to understand that which URs are adopted and how the URs get involved in the posteriors.

- We have clarified the text to highlight that we perform two different analyses: (i) a completely agnostic study, without imposing any assumptions on the matter, i.e. no URs, no requirement that the two objects have the same EoS, (ii) eliminating the f-mode frequency from the GW model so that the matter signatures are characterized solely by Λ using the empirical approximate URs for NSs.

The revised text reads:” In order to constrain fundamental oscillation mode frequencies in GW170817, we re-analyse the publicly available strain data [13, 14] treating the tidal deformabilities Λ_A and the dimensionless angular f-mode frequencies Ω_A of the A-th component object as independent parameters without imposing the aforementioned empirical URs or any requirement that the two compact objects obey the same EoSs.

We then repeat the analysis imposing URs that relate the f - mode frequency and the tidal deformability, $\Omega_A = \sum_i a_i \xi_i$, where $\xi_i = \log(\Lambda)$ and a_i are numerical coefficients[6].”

(3) Please check whether the dimensions of relevant quantities are consistent: in the penultimate paragraph of the right column in page 3, $\Lambda(3,1)$ is [410, 9404] Hz, $\Lambda(3,2)$ is [466, 9446] Hz; while in the last paragraph of the left column in page 5, $\Lambda(\text{prior}; 3,1)$ is dimensionless.

- We have fixed these errors, all quantities are dimensionless.

(4) The symbols of the quantities, such as $\Lambda(2,1)$ and $\Lambda(2,A)$, $f(2,1)$ and $f(2,A)$, $\Lambda(2,A)$ and $\Lambda(2A)$, etc. are easily confused, it is better to explain these symbols in a little more detail.

- We have added an additional note in the text “We recall that the notation for the subscripts $\ell_{2,1}$ is that the first label denotes the multipolar index ℓ , the second specifies the object A with $A=1$ being the larger mass companion.” And in the caption of Figure 1: “... quadrupolar tidal deformability of the larger companion $\Lambda_{2,1}$, where the subscripts denote the multipolar index $\ell=2$ and the larger mass companion $A=1$ ”.

Reviewer #2 (Remarks to the Author):

The authors perform for the first time (and without assuming the so-called universal relations) a Bayesian study on the data of GW170817 to estimate frequencies of the f-modes excited during the final part of the merger. Separately from GW170817, such f-modes had also been studied in previous works, mentioned in the manuscript.

I believe that the results presented in this manuscript are technically sound and expressed in a easy-to-read and convincing manner (actually there are some repetitions, therefore the text could be shortened somewhat, if necessary). For sure these results are of interest to other

researchers in the field, but I do not believe that they represent an advance in understanding likely to influence thinking in the field. In fact, this manuscript shows an application of known techniques and theory to some actual data. This is why I do not recommend publication in Nature

Communications. I would definitely recommend publication in other types of journals.

No comments or suggestions were provided. Though we note that referees 1 and 3 agree on the originality, significance and importance of our work with a recommendation that it should be published.

We would like to take this opportunity to stress the novelty of our work. F-mode studies up to our work have been focused on the post-merger regime, where the hot EoS may be probed. Other works, such as the recent work by Wen et al., on the other hand, used the purely adiabatic information obtained from the observation of GW170817 ($\tilde{\Lambda}$) together with universal relations based on General Relativity to compute the quadrupolar f-mode frequency. We put forward a novel approach to testing the validity of these universal relations by means of simultaneously measuring the tidal deformability and f-mode frequency. The application to GW170817 illustrates the approach while the forecasting for the next generation of ground-based detectors demonstrates the real scientific potential. Our work is foundational: it opens the door to completely novel tests of General Relativity and provides a means to discriminate between neutron stars and exotic compact objects as illustrated in Fig. 1.

Reviewer #3 (Remarks to the Author):

The manuscript deals with a detailed Bayesian analysis of the GW strain data measured on the GW170817 event, by using a model for the GW signal that explicitly depends on both the tidal deformability and the f-mode frequency, assumed as independent variables. Differently from previous analysis, no empirical universal relations (URs) are necessarily assumed and

a first constraint on the f-mode frequency only based on the available data is deduced. Moreover, a simple but clear test is done to reveal how this kind of analysis can improve our understanding of f-modes in future, better resolved detection.

The work is original and has a fairly good significance. Despite the fact that the results obtained are still compatible with the (possibly more constrained) bounds obtained by additionally imposing URs, this analysis has the potential advantage of revealing systematic uncertainties and of testing some of the hypothesis underlying the empirical URs. Moreover, future analysis of this kind, applied to a larger sample of good resolved GW events, can potentially reveal new phases of matter at very large supra-nuclear densities or test alternative theories of gravity.

The results presented can be of interest for a large (even if not extremely large) community. The Authors did an appreciable attempt to put their results in a broader and more understandable contest. However, the work has an intrinsic degree of technicality that cannot be avoided. Still, I do not think that this is a severe limitation of the work.

The data and the methodology are state-of-the-art and the authors are recognized experts in the field of GW data analysis. The references and the provided details are enough to enable reproducing the results. Bayesian analysis strongly depends, among the others, on the priors used in the analysis. The authors have investigated in a satisfactory way the possible dependence of their results on the priors on the f-mode frequency. The study presented in the second part

of the paper is less accurate, but being a proof-of-principle, this is not necessarily a limitation.

The conclusions reflect the main points of the paper. Some of the outlooks are not fully motivated, but it should be possible to amend.

Below I list a few possible suggestions to improve the manuscript:

1) I am not convinced by the switch between the method and the results sections. Some of the questions I had while reading the manuscript were (partially) answered in the method section. So, I am wondering if a more canonical ordering could be better suited.

- We considered structuring the text differently within the constraints provided by the journal format. The current format is the most balanced approach we could find.

2) In a few places in the paper, the Authors speak about ultra-high densities. Since the journal reader target is very broad, I would avoid this too general statements and I would replace them with more precise statement. At which densities do the authors expect their method to reveal matter properties (for example, in units of nuclear saturation density)?

- We have switched to making more precise statements in terms of the nuclear saturation density, as recommended by the referee.

3) The Authors used TaylorF2 waveform models. This is not the only available one, even if it has some advantages. Could the Authors comments on possible pro and cons with respect to other approximants, and if they expect possible differences using different approximants?

- A detailed discussion on waveform systematics is beyond the scope of this paper but some statements on systematics are discussed in the companion model paper (Schmidt et al). Fundamentally, we expect any waveform model with an improved description of the point-particle or adiabatic tidal sector to lead to reduced systematic biases and tighter constraints on the binary parameters (e.g. Samajdar et al, arXiv:1810.03936). A long-term, detailed study to explore aspects of this is underway.

4) When commenting on "the f_l -dependent phase contribution (which is proportional to $\Lambda_{f_l^{-2}}$)", a reference could be useful.

- An explicit reference to Schmidt & Hinderer has now been added.

5) Very minor: sometimes (more often in the second part of the paper) the Authors do not put a comma in subscripts (e.g. Ω_{2A}), while I think it should be done (e.g. $\Omega_{2,A}$).

- We thank the referee for spotting this notational inconsistency. We fully agree and have made the notation self-consistent using the advised notation.

6) Very minor: after eq 3, "we adopt a a low- ..."

- This typo has now been fixed.

7) I do not fully understand why the lower bound on Ω is sometimes 0 and sometimes 0.182 (the lower limit of the URs). Is it not possible to have a single choice? Is there something that I am missing in using two different values?

- The text has been slightly modified to try to clarify this discussion in the paper. The lower limit of the priors is always taken to be $\Omega = 0$. The upper limit on the priors is varied in a range from 0.182 (the black hole limit implied by universal relations) through to 0.5 (far higher than any astrophysically realistic value). The aim of this is to help gauge the sensitivity of our results to the priors and to provide a more complete analysis. The green shaded area in Figure 1 therefore demonstrates how the *lower bound* on the *posteriors* change as we vary the *upper limit* of the *prior* from 0.182 to 0.5. As the SNR in GW170817 is insufficient to overcome the adiabatic plateau, we find that increasing the upper limit on the prior essentially drags the lower bound on the 90% credible regions to higher values of Ω . We hope this clarifies the use of these values and how they relate to the analysis in this paper!

8) When discussing the potential relevance of this kind of analysis in future detection, the Authors generically say that it will be complementary to post-merger analysis. There is however a possible fundamental difference: in the post-merger, the hot nuclear EOS will be probed, while in the inspiral the cold one. Could the Authors expand a little bit on that? Moreover, how does the SNR necessary in the presented analysis relates with the SNR required by post-merger analysis? This could be a relevant term of comparison.

- We have added some text to clarify and reinforce this point as we feel it is of particular interest and importance. The EoS information from the pre-merger phase regards neutron stars that have both a cold EoS but also lighter NS masses. The post-merger phase probes the hot EoS but at relatively larger NS masses. Detailed information on the pre- and post-merger phases will be a very useful as a form of "IMR" (inspiral-merger-ringdown) consistency test for binary neutron stars (or generally exotic compact objects, modified theories of gravity etc). A detailed analysis as to how different constraints in different portions of the EoS parameter space can be meaningfully combined in such an "IMR" consistency test is under development, though naturally far beyond the scope of this paper. We have added some text to highlight this and thank the referee for their comment as combining the insight from the post-merger studies with the novel tests proposed here is a very exciting prospect.

Reviewers' comments:

Reviewer #1 (Remarks to the Author):

The authors have carried out a thorough analysis in response to my concerns and dealt with the issues I had raised. I therefore recommend publication of this manuscript.

Reviewer #3 (Remarks to the Author):

Dear Editor, dear Authors,

I have read the answers to my questions and I am happy with them.

I have only a couple of additional minor points (related with the new text) that I would like the authors to clarify:

1) in the final paragraph of the Discussion, the Authors say:

"...whereas the post-merger phase probes the hot EoS at relatively larger NS masses".

I am a bit confuse by "at relatively larger NS masses". The NS masses are an intrinsic property of the binary. The post merger will probe higher **densities** which, in turn, are possibly verified also in isolated or inspiring more massive NSs (this is even more relevant nowadays, after e.g. GW190425). However, depending on the EOS, the post-merger can probe even larger densities (see for example figures from the recent analysis of Perego, Bernuzzi & Radice EPJA 2019). Maybe replacing "at relatively larger NS masses" with "at densities several times larger than the nuclear saturation density" can remove the ambiguity.

2) A few lines after, "this would enable us to probe the pressure of matter at 1-2 times the nuclear saturation density [42],...". I would have naively expected the relevant density range to be 2-5 times the nuclear saturation density, which are the central densities of cold, merging NS. I don't know if the f-mode are more sensitive to the lower part of the EoS, but that part is already well constraints by other nuclear measurements and by chiral-EFT. So, I think the more relevant constraints that GW astroseismology can provide are possibly more relevant at larger densities. Could the Authors please double check this point?

Moreover, the Science article of Lattimer and Prakash 2004 is certainly very relevant and a good reference, but honestly a bit outdated. What about replacing it with something more recent? For example, there is the more recent review, again from Lattimer 2012 (DOI: 10.1146/annurev-nucl-102711-095018). It is still from the pre-GW era, but at least the maximum NS mass is better taken into account and the focus on the EoS is clearer.

We again thank the referees for their quick response!

With regards to the final two points, we agree with the points raised by referee 3.

1) in the final paragraph of the Discussion, the Authors say:

"...whereas the post-merger phase probes the hot EoS at relatively larger NS masses".

I am a bit confuse by "at relatively larger NS masses". The NS masses are an intrinsic property of the binary. The post merger will probe higher ****densities**** which, in turn, are possibly verified also in isolated or inspiring more massive NSs (this is even more relevant nowadays, after e.g. GW190425). However, depending on the EOS, the post-merger can probe even larger densities (see for example figures from the recent analysis of Perego, Bernuzzi & Radice EPJA 2019). Maybe replacing "at relatively larger NS masses" with "at densities several times larger than the nuclear saturation density" can remove the ambiguity.

- We agree that the new text introduced an ambiguity whereas the intention was to highlight that the post-merger phase can probe even larger densities. We have incorporated the text suggested by the referee to remove any such ambiguity.

2) A few lines after, "this would enable us to probe the pressure of matter at 1-2 times the nuclear saturation density [42],...". I would have naively expected the relevant density range to be 2-5 times the nuclear saturation density, which are the central densities of cold, merging NS. I don't know if the f-mode are more sensitive to the lower part of the EoS, but that part is already well constraints by other nuclear measurements and by chiral-EFT. So, I think the more relevant constraints that GW astroseismology can provide are possibly more relevant at larger densities. Could the Authors please double check this point?

Moreover, the Science article of Lattimer and Prakash 2004 is certainly very relevant and a good reference, but honestly a bit outdated. What about replacing it with something more recent? For example, there is the more recent review, again from Lattimer 2012 (DOI: 10.1146/annurev-nucl-102711-095018). It is still from the pre-GW era, but at least the maximum NS mass is better taken into account and the focus on the EoS is clearer.

- We agree that the reference is outdated and have incorporated the recommended citation to the more recent review article. With regards to the first point, the statement in the paper was somewhat overly conservative. The f-mode frequencies integrate over the entire density range and do probe the EoS at all densities, e.g. phase transitions at high densities can affect the f-modes. However, the explicit dependence has not been systematically quantified and will also depend on the EoS and mass of the NS, with these properties affecting how much of the interior is above ~ 2 x the nuclear saturation density. We have updated the wording to "well above the nuclear saturation density" to reflect this

REVIEWERS' COMMENTS:

Reviewer #3 (Remarks to the Author):

The Authors have addressed the two very minor points.

I have no more additional points and the paper can be accepted on my side.